# Variational Laws of Visual Attention for Dynamic Scenes

**Dario Zanca**
DINFO, University of Florence
DIISM, University of Siena
dario.zanca@unifi.it

**Marco Gori**
DIISM, University of Siena
marco@diism.unisi.it

## Abstract

Computational models of visual attention are at the crossroad of disciplines like cognitive science, computational neuroscience, and computer vision. This paper proposes a model of attentional scanpath that is based on the principle that there are foundational laws that drive the emergence of visual attention. We devise variational laws of the eye-movement that rely on a generalized view of the Least Action Principle in physics. The potential energy captures details as well as peripheral visual features, while the kinetic energy corresponds with the classic interpretation in analytic mechanics. In addition, the Lagrangian contains a brightness invariance term, which characterizes significantly the scanpath trajectories. We obtain differential equations of visual attention as the stationary point of the generalized action, and we propose an algorithm to estimate the model parameters. Finally, we report experimental results to validate the model in tasks of saliency detection.

## 1 Introduction

Eye movements in humans constitute an essential mechanism to disentangle the tremendous amount of information that reaches the retina every second. This mechanism in adults is very sophisticated. In fact, it involves both bottom-up processes, which depend on raw input features, and top-down processes, which include task dependent strategies [2; 3; 4]. It turns out that visual attention is interwound with high level cognitive processes, so as its deep understanding seems to be trapped into a sort of eggs-chicken dilemma. Does visual scene interpretation drive visual attention or the other way around? Which one "was born" first? Interestingly, this dilemma seems to disappears in newborns: despite their lack of knowledge of the world, they exhibit mechanisms of attention to extract relevant information from what they see [5]. Moreover, there are evidences that the very first fixations are highly correlated among adult subjects who are presented with a new input [25]. This shows that they still share a common *mechanism* that drive early fixations, while scanpaths diverge later under top-down influences.

Many attempts have been made in the direction of modeling visual attention. Based on the feature integration theory of attention [14], Koch and Ullman in [9] assume that human attention operates in the early representation, which is basically a set of feature maps. They assume that these maps are then combined in a central representation, namely the *saliency map*, which drives the attention mechanisms. The first complete implementation of this scheme was proposed by Itti *et al.* in [10]. In that paper, feature maps for color, intensity and orientation are extracted at different scales. Then center-surround differences and normalization are computed for each pixel. Finally, all this information is combined linearly in a centralized saliency map. Several other models have been proposed by the computer vision community, in particular to address the problem of refining saliency maps estimation. They usually differ in the definition of saliency, while they postulate a centralized control of the attention mechanism through the saliency map. For instance, it has been claimed that

the attention is driven according to a principle of information maximization [16] or by an opportune selection of surprising regions [17]. A detailed description of the state of the art is given in [8]. Machine learning approaches have been used to learn models of saliency. Judd *et al.* [1] collected 1003 images observed by 15 subjects and trained an SVM classifier with low-, middle-, and high-level features. More recently, automatic feature extraction methods with convolutional neural networks achieved top level performance on saliency estimation [26; 18].

Most of the referred papers share the idea that saliency is the product of a global computation. Some authors also provide scanpaths of image exploration, but to simulate them over the image, they all use the procedure defined by [9]. The *winner-take-all* algorithm is used to select the most salient location for the first fixation. Then three rules are introduced to select the next location: *inhibition-of-return*, *similarity preference,* and *proximity preference*. An attempt of introducing biological biases has been made by [6] to achieve more realistic saccades and improve performance.

In this paper, we present a novel paradigm in which visual attention emerges from a few unifying functional principles. In particular, we assume that attention is driven by the *curiosity* for regions with many details, and by the need to achieve *brightness invariance*, which leads to fixation and motion tracking. These principles are given a mathematical expression by a variational approach based on a generalization of least action, whose stationary point leads to the correspondent Euler-Lagrange differential equations of the focus of attention. The theory herein proposed offers an intriguing model for capturing a mechanisms behind saccadic eye movements, as well as object tracking within the same framework. In order to compare our results with the state of the art in the literature, we have also computed the saliency map by counting the visits in each pixel over a given time window, both on static and dynamic scenes. It is worth mentioning that while many papers rely on models that are purposely designed to optimize the approximation of the saliency map, for the proposed approach such a computation is obtained as a byproduct of a model of scanpath.

The paper is organized as follows. Section 2 provides a mathematical description of the model and the Euler-Lagrange equations of motion that describe attention dynamics. The technical details, including formal derivation of the motion equations, are postponed to the Appendix. In the Section 3 we describe the experimental setup and show performance of the model in a task of saliency detection on two popular dataset of images [12; 11] and one dataset of videos [27]. Some conclusions and critical analysis are finally drawn in Section 4.

## 2 The model

In this section, we propose a model of visual attention that takes place in the earliest stage of vision, which we assume to be completely data driven. We begin discussing the driving principles.

### 2.1 Principles of visual attention

The *brightness* signal $b(t, x)$ can be thought of as a real-valued function

$$b : \mathbb{R}^+ \times \mathbb{R}^2 \to \mathbb{R} \tag{1}$$

where $t$ is the time and $x = (x_1, x_2)$ denotes the position. The *scanpath* over the visual input is defined as

$$x : \mathbb{R}^+ \to \mathbb{R}^2 \tag{2}$$

The scanpath $x(t)$ will be also referred to as *trajectory* or *observation*.

Three fundamental principles drive the model of attention. They lead to the introduction of the correspondent terms of the Lagrangian of the action.

    i) *Boundedness of the trajectory*
        Trajectory $x(t)$ is bounded into a defined area (retina). This is modeled by a harmonic oscillator at the borders of the image which constraints the motion within the retina[1]:

$$V(x) = k \sum_{i=1,2} \left( (l_i - x_i)^2 \cdot [x_i > l_i] + (x_i)^2 \cdot [x_i < 0] \right) \tag{3}$$

where $k$ is the elastic constant, $l_i$ is the i-th dimension of the rectangle which represents the retina[2].

ii) *Curiosity driven principle*

Visual attention is attracted by regions with many details, that is where the magnitude of the gradient of the brightness is high. In addition to this local field, the role of peripheral information is included by processing a blurred version $p(t, x)$ of the brightness $b(t, x)$. The modulation of these two terms is given by

$$C(t, x) = b_x^2 \cos^2(\omega t) + p_x^2 \sin^2(\omega t), \tag{4}$$

where $b_x$ and $p_x$ denote the gradient w.r.t. $x$. Notice that the alternation of the local and peripheral fields has a fundamental role in avoiding trapping into regions with too many details.

iii) *brightness invariance*

Trajectories that exhibit *brightness invariance* are motivated by the need to perform fixation. Formally, we impose the constraint $\dot{b} = b_t + b_x \dot{x} = 0$. This is in fact the classic constraint that is widely used in computer vision for the estimation of the optical flow [20]. Its soft-satisfaction can be expressed by the associated term

$$B(t, x, \dot{x}) = \left(b_t + b_x \dot{x}\right)^2. \tag{5}$$

Notice that, in the case of static images, $b_t = 0$, and the term is fully satisfied for trajectory $x(t)$ whose velocity $\dot{x}$ is perpendicular to the gradient, *i.e.* when the focus is on the borders of the objects. This kind of behavior favors coherent fixation of objects. Interestingly, in case of static images, the model can conveniently be simplified by using the upper bound of the brightness as follows:

$$B(t, x, \dot{x}) = \dot{b}^2(t, x) = (\partial b_t + b_x \dot{x})^2 \leq$$
$$\leq 2b_t^2 + 2b_x^2 \dot{x}^2 := \bar{B}(t, x, \dot{x}) \tag{6}$$

This inequality comes from the parallelogram law of Hilbert spaces. As it will be seen the rest of the paper, this approximation significantly simplifies the motion equations.

## 2.2 Least Action Principle

Visual attention scanpaths are modeled as the motion of a particle of mass $m$ within a potential field. This makes it possible to construct the generalized action

$$S = \int_0^T L(t, x, \dot{x}) \, dt \tag{7}$$

where $L = K - U$, where $K$ is the kinetic energy

$$K(\dot{x}) = \frac{1}{2} m \dot{x}^2 \tag{8}$$

and $U$ is a generalized potential energy defined as

$$U(t, x, \dot{x}) = V(x) - \eta C(t, x) + \lambda B(t, x, \dot{x}). \tag{9}$$

Here, we assume that $\eta, \lambda > 0$. Notice, in passing that while $V$ and $B$ get the usual sign of potentials, $C$ comes with the flipped sign. This is due to the fact that, whenever it is large, it generates an attractive field. In addition, we notice that the brightness invariance term is not a truly potential, since it depends on both the position and the velocity. However, its generalized interpretation as a "potential" comes from considering that it generates a force field. In order to discover the trajectory we look for a stationary point of the action in Eq. (7), which corresponds to the Euler-Lagrange equations

$$\frac{d}{dt} \frac{\partial L}{\partial \dot{x}_i} = \frac{\partial L}{\partial x_i}, \tag{10}$$

where $i = 1, 2$ for the two motion coordinates. The right-hand term in (10) can be written as

$$\frac{\partial L}{\partial x} = \eta C_x - V_x - \lambda B_x. \tag{11}$$

Likewise we have

$$\frac{d}{dt}\frac{\partial L}{\partial \dot{x}} = m\ddot{x} - \lambda\frac{d}{dt}B_{\dot{x}} \tag{12}$$

so as the general motion equation turns out to be

$$m\ddot{x} - \lambda\frac{d}{dt}B_{\dot{x}} + V_x - \eta C_x + \lambda B_x = 0. \tag{13}$$

These are the general equations of visual attention. In the Appendix we give the technical details of the derivations. Throughout the paper, the proposed model is referred to as the EYe MOvement Laws (EYMOL).

## 2.3 Parameters estimation with simulated annealing

Different choices of parameters lead to different behaviors of the system. In particular, weights can emphasize the contribution of curiosity or brightness invariance terms. To better control the system we use two different parameters for the curiosity term, namely $\eta_b$ and $\eta_p$, to weight $b$ and $p$ contributions respectively. The best values for the three parameters $(\eta_b, \eta_p, \lambda)$ are estimated using the algorithm of simulated annealing (*SA*). This method allows to perform iterative improvements, starting from a known state $i$. At each step, the SA considers some neighbouring state $j$ of the current state, and probabilistically moves to the new state $j$ or stays on the current state $i$. For our specific problem, we limit our search to a parallelepiped-domain $D$ of possible values, due to theoretical bounds and numerical[3] issues. Distance between states $i$ and $j$ is proportional with a temperature $T$, which is initialized to 1 and decreases over time as $T_k = \alpha * T_{k-1}$, where $k$ identifies the iteration step, and $0 << \alpha < 1$. The iteration step is repeated until the system reaches a state that is good enough for the application, which in our case is to maximize the NSS similarity between human saliency maps and simulated saliency maps.
Only a batch of a 100 images from CAT2000-TRAIN is used to perform the *SA* algorithm[4]. This batch is created by randomly selecting 5 images from each of the 20 categories of the dataset. To start the *SA*, parameters are initialized in the middle point of the 3-dimensional parameters domain $D$. The process is repeated 5 times, on different sub-samples, to select 5 parameters configurations. Finally, those configurations together with the average configuration are tested on the whole dataset, to select the best one.

---

**Algorithm 1** In the psedo-code, P() is the acceptance probability and score() is computed as the average of NSS scores on the sample batch of 100 images.

---

```
 1: procedure SIMULATEDANNEALING
 2:     Select an initial state i ∈ D
 3:     T ← 1
 4:     do
 5:         Generate random state j, neighbor of i
 6:         if  P(score(i), score(j)) ≥ Random(0, 1) then
 7:             i ← j
 8:         end if
 9:         T ← α * T
10:     while T ≥ 0.01
11: end procedure
```

| Model version | MIT1003 | | CAT2000-TRAIN | |
|---|---|---|---|---|
| | AUC | NSS | AUC | NSS |
| V1 (approx. br. inv.) | 0.7996 (0.0002) | 1.2784 (0.0003) | 0.8393 (0.0001) | 1.8208 (0.0015) |
| V2 (exact br. inv.) | 0.7990 (0.0003) | 1.2865 (0.0039) | 0.8376 (0.0013) | 1.8103 (0.0137) |

Table 1: Results on MIT1003 [1] and CAT2000-TRAIN [11] of the two different version of EYMOL. Between brackets is indicated the standard error.

## 3    Experiments

To quantitative evaluate how well our model predicts human fixations, we defined an experimental setup for salient detection both in images and in video. We used images from MIT1003 [1], MIT300 [12] and CAT2000 [11], and video from SFU [27] eye-tracking database. Many of the design choices were common to both experiments; when they differ, it is explicitly specified.

### 3.1    Input pre-processing

All input images are converted to gray-scale. Peripheral input $p$ is implemented as a blurred versions of the brightness $b$. This blurred version is obtained by convolving the original gray-scale image with a Gaussian kernel. For the images only, an algorithm identifies the rectangular zone of the input image in which the totality of information is contained in order to compute $l_i$ in (14). Finally both $b$ and $p$ are multiplied by a Gaussian blob centered in the middle of the frame in order to make brightness gradients smaller as we move toward periphery and produce a center bias.

### 3.2    Saliency maps computation

Differently by many of the most popular methodologies in the state-of-the-art [10; 16; 1; 24; 18], the saliency map is not itself the central component of our model but it can be naturally calculated from the visual attention laws in (13). The output of the model is a trajectory determined by a system of two second ordered differential equations, provided with a set of initial conditions. Since numerical integration of (13) does not raise big numerical difficulties, we used standard functions of the python scientific library *SciPy* [21].

Saliency map is then calculated by summing up the most visited locations during a sufficiently large number of virtual observations. For images, we collected data by running the model 199 times, each run was randomly initialized almost at the center of the image and with a small random velocity, and integrated for a running time corresponding to 1 second of visual exploration. For videos, we collected data by running the model 100 times, each run was initialized almost at the center of the first frame of the clip and with a small random velocity.

Model that have some blur and center bias on the saliency map can improve their score with respect to some metrics. A grid search over *blur radius* and *center* parameter $\sigma$ have been used, in order to maximize AUC-Judd and NSS score on the training data of CAT2000 in the case of images, and on SFU in case of videos.

### 3.3    Saliency detection on images

Two versions of the the model have been evaluated. The first version V1 implementing brightness invariance in the approximated form (6), the second version V2 implementing the brightness invariance in its exact form, as described in the Appendix. Model V1 and V2 have been compared on the MIT1003 and CAT2000-TRAIN datasets, since they provide public data about fixations. Parameters estimation have been conducted independently for the two models and the best configuration for each one is used in this comparison. Results are statistically equivalent (see Table2) and this proves that, in the case of static images, the approximation is very good and does not cause loss in the score. For further experiments we decided to use the approximated form V1 due to its simpler form of the equation that also reduces time of computation.

Model V1 has been evaluated in two different dataset of eye-tracking data: MIT300 and CAT2000-TEST. In this case, scores were officially provided by MIT Saliency Benchmark Team [15]. Description of the metrics used is provided in [13]. Table 2 and Table 3 shows the scores of our

| | MIT300 | | | | | |
|---|---|---|---|---|---|---|
| | AUC | SIM | EMD | CC | NSS | KL |
| Itti-Koch [10], implem. by [19] | 0.75 | 0.44 | 4.26 | 0.37 | 0.97 | **1.03** |
| AIM [16] | 0.77 | 0.40 | 4.73 | 0.31 | 0.79 | 1.18 |
| Judd Model [1] | 0.81 | 0.42 | 4.45 | **0.47** | **1.18** | 1.12 |
| AWS [24] | 0.74 | 0.43 | 4.62 | 0.37 | 1.01 | 1.07 |
| eDN [18] | **0.82** | 0.44 | 4.56 | 0.45 | 1.14 | 1.14 |
| **EYMOL** | 0.77 | **0.46** | **3.64** | 0.43 | 1.06 | 1.53 |

Table 2: Results on MIT300 [12] provided by MIT Saliency Benchmark Team [15]. The models are sorted chronologically. In bold, the best results for each metric and benchmarks.

| | CAT2000-TEST | | | | | |
|---|---|---|---|---|---|---|
| | AUC | SIM | EMD | CC | NSS | KL |
| Itti-Koch [10], implem. by [19] | 0.77 | 0.48 | 3.44 | 0.42 | 1.06 | **0.92** |
| AIM [16] | 0.76 | 0.44 | 3.69 | 0.36 | 0.89 | 1.13 |
| Judd Model [1] | 0.84 | 0.46 | 3.60 | 0.54 | 1.30 | 0.94 |
| AWS [24] | 0.76 | 0.49 | 3.36 | 0.42 | 1.09 | 0.94 |
| eDN [18] | **0.85** | 0.52 | 2.64 | 0.54 | 1.30 | 0.97 |
| **EYMOL** | 0.83 | **0.61** | **1.91** | **0.72** | **1.78** | 1.67 |

Table 3: Results on CAT2000 [11] provided by MIT Saliency Benchmark Team [15]. The models are sorted chronologically. In bold, the best results for each metric and benchmarks.

model compared with five other popular method [10; 16; 1; 24; 18], which have been selected to be representative of different approaches. Despite its simplicity, our model reaches best score in half of the cases and for different metrics.

### 3.4 Saliency detection on dynamic scenes

We evaluated our model in a task of saliency detection with the dataset SFU [27]. The dataset contains 12 clips and fixations of 15 observers, each of them have watched twice every video. Table 4 provides a comparison with other four model. Also in this case, despite of its simplicity and even if it was not designed for the specific task, our model competes well with state-of-the-art models. Our model can be easily run in real-time to produce an attentive scanpath. In some favorable case, it shows evidences of tracking moving objects on the scene.

| | SFU Eye-Tracking Database | | | | |
|---|---|---|---|---|---|
| | EYMOL | Itti-Koch [10] | Surprise [17] | Judd Model [1] | HEVC [28] |
| Mean AUC | 0.817 | 0.70 | 0.66 | 0.77 | 0.83 |
| Mean NSS | 1.015 | 0.28 | 0.48 | 1.06 | 1.41 |

Table 4: Results on the video dataset SFU [27]. Scores are calculated as the mean of AUC and NSS metrics of all frames of each clip, and then averaged for the 12 clips.

## 4 Conclusions

In this paper we investigated how human attention mechanisms emerge in the early stage of vision, which we assume completely data-driven. The proposed model consists of differential equations, which provide a real-time model of scanpath. These equations are derived in a generalized framework of least action, which nicely resembles related derivations of laws in physics. A remarkable novelty concerns the unified interpretation of curiosity-driven movements and the brightness invariance term for fixation and tracking, that are regarded as mechanisms that jointly contribute to optimize the acquisition of visual information. Experimental results on both image and video datasets of saliency are very promising, especially if we consider that the proposed theory offers a truly model of eye movements, whereas the computation of the saliency maps only arises as a byproduct.

In future work, we intend to investigate behavioural data, not only in terms of saliency maps, but also by comparing actual generated scanpaths with human data in order to discover temporal correlations. We aim at providing the integration of the presented model with a theory of feature extraction that is still expressed in terms of variational-based laws of learning [29].

## Appendix: Euler-Lagrange equations

In this section we explicitly compute the differential laws of visual attention that describe the visual attention scanpath, as the Euler-Lagrange equations of the action functional (7).

First, we compute the partial derivatives of the different contributions w.r.t. $x$, in order to compute the exact contributions of (11). For the retina boundaries,

$$V_x = k \sum_{i=1,2} \left( -2\left(l_i - x_i\right) \cdot [x_i > l_i] + 2x_i \cdot [x_i < 0] \right) \tag{14}$$

The curiosity term (4)

$$C_x = 2cos^2(\omega t)b_x \cdot b_{xx} + 2sin^2(\omega t)p_x \cdot p_{xx} \tag{15}$$

For the term of brightness invariance,

$$B_x = \frac{\partial}{\partial x}\left(b_t + b_x \dot{x}\right)^2 \tag{16}$$

$$= 2\left(b_t + b_x \dot{x}\right)\left(b_{tx} + b_{xx}\dot{x}\right) \tag{17}$$

Since we assume $b \in \mathcal{C}^2(t,x)$, by the Schwarz's theorem[5], we have that $b_{tx} = b_{xt}$, so that

$$B_x = 2\left(b_t + b_x \dot{x}\right)\left(b_{xt} + b_{xx}\dot{x}\right) \tag{18}$$

$$= 2(\dot{b})(\dot{b}_x) \tag{19}$$

We proceed by computing the contribution in (12). Derivative w.r.t. $\dot{x}$ of the brightness invariance term is

$$B_{\dot{x}} = \frac{\partial}{\partial \dot{x}}\left(b_t + b_x \dot{x}\right)^2 \tag{20}$$

$$= 2\left(b_t + b_x \dot{x}\right)b_x \tag{21}$$

$$= 2(\dot{b})(b_x) \tag{22}$$

So that, total derivative w.r.t. $t$ can be write as

$$\frac{d}{dt}B_{\dot{x}} = 2\left(\ddot{b}b_x + \dot{b}\dot{b}_x\right) \tag{23}$$

We observe that $\ddot{b} \equiv \ddot{b}(t,x,\dot{x},\ddot{x})$ is the only term which depends on second derivatives of $x$. Since we are interested in expressing EL in an explicit form for the variable $\ddot{x}$, we explore more closely its contribution

$$\ddot{b}(t,x,\dot{x},\ddot{x}) = \frac{d}{dt}\dot{b} \tag{24}$$

$$= \frac{d}{dt}(b_t + b_x \dot{x}) \tag{25}$$

$$= \dot{b}_t + \dot{b}_x \cdot \dot{x} + b_x \cdot \ddot{x} \tag{26}$$

$$\tag{27}$$

Substituting it in (23) we have

$$\frac{d}{dt}B_{\dot{x}} = 2\left((\dot{b}_t + \dot{b}_x \cdot \dot{x} + b_x \cdot \ddot{x})b_x + \dot{b}\dot{b}_x\right) \tag{28}$$

$$= 2\left((\dot{b}_t + \dot{b}_x \cdot \dot{x})b_x + \dot{b}\dot{b}_x\right) + 2(b_x \cdot \ddot{x})b_x \tag{29}$$

So that, from (12) we get

$$\frac{d}{dt}\frac{\partial L}{\partial \dot{x}} = m\ddot{x} - 2\lambda\Big((\dot{b}_t + \dot{b}_x \cdot \dot{x})b_x + \dot{b}\dot{b}_x + (b_x \cdot \ddot{x})b_x\Big) \tag{30}$$

**Euler-Lagrange equations.** Combining (11) and (30), we get Euler-Lagrange equation of attention

$$m\ddot{x} - 2\lambda\Big((\dot{b}_t + \dot{b}_x \cdot \dot{x})(b_x) + (\dot{b})(\dot{b}_x) + (b_x \cdot \ddot{x})b_x\Big) = \eta C_x - V_x - \lambda B_x \tag{31}$$

In order to obtain explicit form for the variable $\ddot{x}$, we re-write the equation as to move to the left all contributes which do not depend on that variable.

$$m\ddot{x} - 2\lambda(b_x \cdot \ddot{x})b_x = \eta C_x - V_x - \lambda B_x + 2\lambda((\dot{b}_t + \dot{b}_x \cdot \dot{x})(b_x) + (\dot{b})(\dot{b}_x)) \tag{32}$$

$$= \underbrace{\eta C_x - V_x + 2\lambda(\dot{b}_t + \dot{b}_x \cdot \dot{x})(b_x)}_{A=(A_1, A_2)} \tag{33}$$

In matrix form, the equation is

$$\begin{pmatrix} m\ddot{x}_1 \\ m\ddot{x}_2 \end{pmatrix} - \begin{pmatrix} 2\lambda(b_{x_1}\ddot{x}_1 + b_{x_2}\ddot{x}_2)b_{x_1} \\ 2\lambda(b_{x_1}\ddot{x}_1 + b_{x_2}\ddot{x}_2)b_{x_2} \end{pmatrix} = \begin{pmatrix} A_1 \\ A_2 \end{pmatrix} \tag{34}$$

which gives us the system of two differential equations

$$\begin{cases} m\ddot{x}_1 - 2\lambda(b_{x_1}\ddot{x}_1 + b_{x_2}\ddot{x}_2)b_{x_1} = A_1 \\ m\ddot{x}_2 - 2\lambda(b_{x_1}\ddot{x}_1 + b_{x_2}\ddot{x}_2)b_{x_2} = A_2 \end{cases} \tag{35}$$

Grouping by same variable,

$$\begin{cases} (m - 2\lambda b_{x_1}^2)\ddot{x}_1 - 2\lambda(b_{x_1}b_{x_2})\ddot{x}_2 & = A_1 \\ -2\lambda(b_{x_1}b_{x_2})\ddot{x}_1 + (m - 2\lambda b_{x_2}^2)\ddot{x}_2 & = A_2 \end{cases} \tag{36}$$

We define

$$D = \begin{vmatrix} (m - 2\lambda b_{x_1}^2) & -2\lambda(b_{x_1}b_{x_2}) \\ -2\lambda(b_{x_1}b_{x_2}) & (m - 2\lambda b_{x_2}^2) \end{vmatrix} \tag{37}$$

$$D_1 = \begin{vmatrix} A_1 & -2\lambda(b_{x_1}b_{x_2}) \\ A_2 & (m - 2\lambda b_{x_2}^2) \end{vmatrix}, D_2 = \begin{vmatrix} (m - 2\lambda b_{x_1}^2) & A_1 \\ -2\lambda(b_{x_1}b_{x_2}) & A_2 \end{vmatrix} \tag{38}$$

By the Cramer's method we get differential equation of visual attention for the two spatial component, *i.e.*

$$\begin{cases} \ddot{x}_1 = \dfrac{D_1}{D} \\ \\ \ddot{x}_2 = \dfrac{D_2}{D} \end{cases} \tag{39}$$

Notice that, this raise to a further condition over the parameter $\lambda$. In particular, in the case values of $b(t,x)$ are normalized in the range $[0,1]$, it imposes to chose

$$D \neq 0 \implies \lambda < \frac{m}{4} \tag{40}$$

In fact,

$$D = (m - 2\lambda b_{x_1}^2)(m - 2\lambda b_{x_2}^2) - 4\lambda^2(b_{x_1}b_{x_2})^2 \tag{41}$$

$$= m\Big(m - 2\lambda(b_{x_1}^2 + b_{x_1}^2)\Big) \tag{42}$$

For values of $b_x = 0$, we have that

$$D = m^2 > 0 \tag{43}$$

so that $\forall t$, we must impose

$$D > 0. \tag{44}$$

If $\lambda > 0$, then

$$m - 2\lambda(b_{x_1}^2 + b_{x_1}^2) > 0 \tag{45}$$

$$\lambda < \frac{m}{2(b_{x_1}^2 + b_{x_1}^2)} \tag{46}$$

The quantity on the right reaches its minimum at $\dfrac{m}{4}$, so that the condition

$$0 < \lambda < \frac{m}{4} \tag{47}$$

guarantees the well-posedness of the problem.

## Footnotes

[1] Here, we use Iverson's notation, according to which if $p$ is a proposition then $[p] = 1$ if $p$=`true` and $[p] = 0$ otherwise

[2] A straightforward extension can be given for circular retina.

[3]Too high values for $\eta_b$ or $\eta_p$ produce numerically unstable and unrealistic trajectories for the focus of attention.

[4]Each step of the *SA* algorithm needs evaluation over all the selected images. Considering the whole dataset would be very expensive in terms of time.

[5]Schwarz's theorem states that, if $f : \mathbb{R}^n \to \mathbb{R}$ has continuous second partial derivatives at any given point in $\mathbb{R}^n$, then $\forall i,j \in \{1,...,n\}$ it holds $f_{x_i x_j} = f_{x_j x_i}$

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
