[Reviews · NeurIPS 2017]

Reviewer 1



This computational neuroscience paper proposes a bottom-up visual feature saliency model combined modified by dynamical systems modelling that aims at capturing the dynamics of scanpaths. I enjoyed reading this paper, it has a set of nice of ideas in defining "affordances" (curiosity, brightness invariance) that drive the scan-path, as well as the idea of the Least Action Principle. These "affordances" are albeit perhaps a bit adhoc postulated and not further motivated. In the context of the Least Action Principle and its derivation in this context it would have been interesting to related it to actual work in eye movement research such as past and recent work by the Wolpert lab. While the lack of biological motivation is in itself alone a minor issue in the era of benchmark-busting deep learning models of (static) visual salience, the issue with this model is the mixed performance it achieves with respect to these models is poor. Given that the model claims to capture biological eye movement dynamics, it would have been intersting to see in the main paper actual figures characterising the generated scan paths, not only in terms of their overall statics at matching eye movem

Reviewer 2



Variational Laws of Visual Attention for Dynamic Scenes The authors investigate what locations in static and dynamic images tend to be attended to by humans. They derive a model by first defining three basic principles for visual attention (defined as an energy function to be minimized by the movements of the eye): (1) Eye movements are constrained by a harmonic oscillator at the borders of the image within a limited-sized retina. (2) a “curiosity driven principle” highlighting the regions with large changes in brightness in both a fine and blurred version of the image, and (3) brightness invariance, which increases as a function of changes in brightness. Using a cost function derived from these three functions, the authors derive differential equations for predicting the eye movements across static or dynamic images (depending on the starting location and initial velocity). The authors evaluate their technique quantitatively on data sets of static and dynamic scenes coupled with human eye movements. They demonstrate that their method performs comparable to the state-of-the-art. Formal definitions of saliency and modeling eye movements are critical issues in computational vision and cognitive science. Psychologists have long been plagued by vague definitions of saliency, and the authors propose a novel and innovative model (as far as I am aware) that could aid the development of better understanding how what makes something salient and a formal model for eye movements (within the bottom-up tradition). Although it is not necessarily state-of-the-art on every metric for every data set, it performs well and provides a refreshingly different perspective on the problem. Unfortunately, some of what I wrote above is based on conjecture as the paper is poorly written and hard to follow. I recommend the authors have others proofread the paper and expand on abbreviations (both within the equations and also those used in Section 3). I would recommend they move the 2 page Appendix to supplementary material and use those extra pages to define each variable and function used (even if it is a convention within your own field – the NIPS audience comes from many different disciplines and some will have trouble following the mathematics otherwise). As a psychologist, I would have liked the authors to connect their work to some of the psychological literature on eye movements as optimal steps for gathering information. See for example, Najemnik, J., & Geisler, W. S. (2008). Eye movement statistics in humans are consistent with an optimal search strategy. Journal of Vision, 8(3), 1-14. There are other relevant articles (particularly from Geisler’s lab, but that should give the authors a pointer to follow into that literature). I would be interested to see a discussion of how their approach compares to their findings (e.g., are they restatements of similar models or provide independent information that could be integrated to produce a better model?)

Reviewer 3



The paper proposes a new approach to the study of eye movements. The authors correctly summarize the current state of the art (as far as I understand it, but I admit from the outset I am not an expert in perception or eye movements). Basically these approaches are based on a saliency map that is defined over an image and fixations points are selected in a optimization/maximization approach. This model (EYMOL) works according to a different assumption and instead is defined directly on trajectories of gaze according to a "Least Action Principle." The details of some of the math were beyond my ability to evaluate because I don't have the necessary physics background (particularly the extensive appendix). However, even still I was able to understand the key elements of the approach and how it differs from past models. The model is helpfully applied to real data set of eye-movements and is compared against a variety of alternative models. While it doesn't dominate on every measure, the results are favorable for the new approach and demonstrate that it has some validity. I think the paper might be of interest to some people in the vision science community (e.g., attendees of VSS). I'm not quite as convinced this makes a great contribution to NIPS, however I think that should be evaluated against other reviewer's opinions who are more expert. I think it likely advances a new and interesting theory that could inspire further empirical research, and so has both novelty and merit for perception sciences.